# REVISITING BFLOAT16 TRAINING

## ABSTRACT

State-of-the-art generic low-precision training algorithms use a mix of 16-bit and 32-bit precision, creating the folklore that 16-bit precision alone is not enough to maximize model accuracy. As a result, deep learning accelerators are forced to support both 16-bit and 32-bit compute units which is more costly than only using 16-bit units for hardware design. We ask *can we do pure 16-bit training which requires only 16-bit compute units, while still matching the model accuracy attained by 32-bit training*. Towards this end, we study pure 16-bit training algorithms on the widely adopted BFloat16 compute unit. While these units conventionally use *nearest rounding* to cast output to 16-bit precision, we show that nearest rounding for model weight updates can often cancel small updates, which degrades the convergence and model accuracy. Motivated by this, we identify two simple existing techniques, stochastic rounding and Kahan summation, to remedy the model accuracy degradation in pure 16-bit training. We empirically show that these two techniques can enable up to 7% absolute validation accuracy gain in pure 16-bit training. This leads to 0.1% lower to 0.2% higher matching validation accuracy compared to 32-bit precision training across seven deep learning applications.

## 1 INTRODUCTION

Recently there has been an explosion in the compute resources required for training deep learning models (Shoeybi et al., 2019; Rajbhandari et al., 2019; Real et al., 2019). As a result, there has been broad interest in leveraging low-precision ($<$ 32-bit) training algorithms to reduce the required compute resources (De Sa et al., 2017; Hubara et al., 2017; Gupta et al., 2015). Among these algorithms, mixed-precision training—in which model activations and gradients are stored using a 16-bit floating point format while model weights and optimizer states use 32-bit precision—is commonly used when training generic deep learning models (Micikevicius et al., 2017; Kalamkar et al., 2019). While there is a wide body of literature showing that low-precision training can minimally impact accuracy on specific models (Wang et al., 2018b; De Sa et al., 2015; Zhang et al., 2017), conventional wisdom suggests that at least some 32-bit computation is required as a fail-safe in generic deep learning training. As such, new accelerator architectures for deep learning are forced to support both 32-bit and 16-bit compute units. This is much more costly in terms of area, power, and speed when compared to hardware with only 16-bit compute units (Horowitz, 2014; Galal et al., 2013).

In this paper we question if 32-bit compute units are truly needed for new deep learning hardware accelerators. Namely, can we match the model accuracy of 32-bit-precision algorithms while leveraging *only* 16-bit compute units? To answer this question, we study *pure 16-bit training* algorithms, ones which use only 16-bit compute units and which store activations, gradients, model weights, and optimizer states all in a 16-bit precision. Specifically, we focus on training with the BFloat16 compute unit which is widely adopted in modern deep learning accelerators (Jouppi et al., 2017; Burgess et al., 2019). Such units take 16-bit inputs, perform computation, and then round the results to a 16-bit output. BFloat16 compute units can provide $3\times$ higher power efficiency, $1.5\times$ lower latency, and $1.5\times$ less chip area than 32-bit units (Horowitz, 2014; Galal et al., 2013). In addition, pure 16-bit training algorithms can reduce the memory footprint and bandwidth consumption of model weights and optimizers by $2\times$ compared to mixed precision or 32-bit precision training, especially for large models with billions of weights (Shoeybi et al., 2019; Rajbhandari et al., 2019). Developing reliable pure 16-bit training algorithms will enable hardware designers to realize these advantages.

The simplest approach to pure 16-bit training is to take a 32-bit baseline and "make it low-precision" by replacing all the 32-bit numbers with 16-bit numbers and replacing each 32-bit floating-point op-

eration with its 16-bit analog, using nearest rounding[1] to quantize as necessary: we call this approach the *standard* algorithm. *Unfortunately, we show empirically that standard pure 16-bit training does not match 32-bit training on model accuracy across deep learning models.* For example, the standard pure 16-bit training algorithm one would run on conventional hardware attains $16\%$ and $7\%$ lower training and validation accuracies than a 32-bit baseline. Motivated by this observation, we start by analyzing what factors limit the model accuracy of this standard pure 16-bit algorithm.

The goal of our analysis is to inspire a clean, minimal set of simple techniques that allow pure 16-bit training to attain strong model accuracy for state-of-the-art deep learning models across application domains. Towards this end, we derive insights from a simple least-squares regression model in Section 3. Using this least-squares regression model, we reveal that nearest rounding of compute unit outputs causes significant convergence degradation and consequent model accuracy loss. More concretely, we show a key theoretical insight hidden in existing work: when running stochastic gradient descent on a least-squares regression model, nearest rounding while updating model weights ignores small updates. This phenomenon significantly degrades the convergence of stochastic gradient descent when model updates become small relative to model weights, which is also what we observe when training deep learning models. In comparison, nearest rounding in the forward and backward pass of backpropagation has a negligible impact on convergence. These insights lead us to consider two simple existing techniques to achieve high-accuracy pure 16-bit training. First, we can use *stochastic rounding* instead of nearest rounding for the model weight updates. Here, the rounded weights become an unbiased estimate of the precise weights without rounding: thus, regardless of the magnitude of updates, the expectation of rounded weights converges at the same speed as the precise weights. Second, we can use the well-known Kahan summation algorithm (Kahan, 1965) to accumulate model updates while still keeping nearest rounding for all operations. This method tracks and compensates weight rounding errors across iterations with auxiliary 16-bit values, which avoids catastrophic cancellation of many small model weight updates.

Empirically, in Section 4 we first validate that, as suggested by our theory, nearest rounding for model weight updates is the sole bottleneck for convergence and model accuracy on several deep learning models. We then demonstrate that pure 16-bit training using stochastic rounding or Kahan summation on model weight updates can match 32-bit training in model accuracy across a wide range of applications (He et al., 2016; Amodei et al., 2016; Devlin et al., 2018; Naumov et al., 2019). To validate that nearest rounding for model weight updates is the cause of the accuracy degradation, we show that if we store model weights in 32-bit precision without rounding during weight updates, and we keep using 16-bits and nearest rounding for all other operations, then the attained model accuracy matches full 32-bit precision training. Next, we demonstrate that 16-bit training with stochastic rounding for weight updates attains model accuracy matching 32-bit training for five out of seven applications in our study. Note that while it works most of the time, this is not a silver bullet, as using stochastic rounding alone could not fully match 32-bit training on all models. To address this, we show that Kahan summation for model weight updates closes remaining gaps on all the models we consider; this Kahan summation comes with a trade off, as it requires $2\times$ weight memory, but achieves up to $0.2\%$ higher validation accuracy than stochastic rounding. *Our results suggest that deep learning accelerators using only 16-bit compute units are feasible if stochastic rounding and Kahan summation are supported respectively by the hardware and the software stack.*

## 2 PRELIMINARY

In this section we establish the background and notation for our study and present the preliminary observations that motivate our work. We focus on the case of stochastic gradient descent (SGD), which is the primary workhorse used to train deep learning models. SGD computes gradients from a subset of training samples, and uses them to update the model weights so as to decrease the loss in expectation. In the classic supervised learning setting, let $(\boldsymbol{X}, \boldsymbol{y})$ be a dataset where $\boldsymbol{X} = [\boldsymbol{x}_1, \boldsymbol{x}_2, ..., \boldsymbol{x}_n] \in \mathbb{R}^{n \times d}$ and $\boldsymbol{y} = (y_1, y_2, ..., y_n) \in \mathbb{R}^n$. On this dataset, we use stochastic gradient descent to optimize a loss function $f(\boldsymbol{w}) = 1/n \sum_{i=1}^{n} f_i(\boldsymbol{w}, \boldsymbol{x}_i, y_i)$ defined by the model. At the $t$-th iteration, we sample an index subset $\sigma(t) \subset \{1, 2, .., n\}$ and compute a sample gradient $\nabla f_{\sigma(t)}(\boldsymbol{w}_t)$ as an unbiased estimate of the full gradient $\nabla f(\boldsymbol{w})$. In deep learning, model training can be described as a compute graph where the compute graph operators such as addition and ma-

---

[1]This nearest rounding is the standard rounding mode for compute unit output commonly supported across hardware platforms (Intel, 2018; Nvidia, 2020).

trix multiplication are the nodes. For example, the model weight update operator is defined as the subtraction in the operation $\boldsymbol{w}_{t+1} = \boldsymbol{w}_t - \alpha \nabla f_{\sigma(t)}(\boldsymbol{w}_t)$ which updates the model weight $\boldsymbol{w}$.

**16-bit Compute Units**   On modern hardware, numerical operators in the compute graph are supported by fused multiply-and-accumulation (FMAC) compute units. These units work by computing $a \leftarrow a + (x \times y)$, where $x$ and $y$ are input floating point numbers, and $a$ is an accumulator that is part of the FMAC unit. Importantly, for a 16-bit FMAC unit, the accumulator $a$ has higher-than-16-bit precision. This higher precision accumulator is inexpensive compared to the multiply in terms of chip area and energy consumption in FMAC units, but is critical to the numerical accuracy of operations such as matrix multiplication and convolution. Thus 16-bit FMAC units with higher precision accumulator is standard for modern hardware including TPUs and GPUs (Chao & Saeta, 2019; Markidis et al., 2018; Stephens, 2019); this will likely continue to be standard in emerging accelerators. Because of the higher precision accumulator, the result in the accumulator then needs to be *rounded* to 16-bits before it is output from the FMAC unit (e.g. before writing to memory). FMAC units use the same hardware implementation to support all operators from simple additions to computationally intensive convolutions, so this output-rounding step happens for all the operators in a compute graph.

**Nearest Rounding**   FMAC output rounding is widely implemented with *nearest rounding* as the standard mode, due to its efficient support across hardware platforms. "Nearest rounding" means rounding a higher precision floating point number to the closest low-precision representable value. *Given that the add step already uses accurate higher preicision accumulators, this nearest rounding is the primary source of numerical errors for training using 16-bit FMAC units.* In this context of 16 bit FMAC units and nearest rounding, we discuss the following training-precision approaches.

- In *32-bit precision training*, all the compute graph operators read and write memory using a 32-bit precision. These operators require 32-bit compute units, which constrains the compute and memory efficiency.

- In *mixed precision training*, model weights are stored in 32-bit precision while activations and gradients use 16-bit precision. Thus, new accelerators customized to maximize efficiency for mixed precision training still require 32-bit compute units for operators involving 32-bit weights as the input; this has lower efficiency in power, speed and chip area than only using 16-bit units.

- In *pure 16-bit training*, activation, gradients and model weights are all stored in 16-bit precision. All operators use pure 16-bit input and write out 16-bit output after rounding. Thus, aside from just saving memory, pure 16-bit training can eliminate the requirement for 32-bit compute units. This opens the possibility for highly-efficient accelerators using only 16-bit compute units. In spite of this favorable efficiency, we now show that it can be surprisingly challenging for standard pure 16-bit training to match 32-bit training on model accuracy.

**Motivating Observations**   Although recent works have shown that certain models are robust to numerical error during training (Wang et al., 2018b; De Sa et al., 2015; Zhang et al., 2017), surprisingly, we observe that it is challenging for pure 16-bit training to attain the same accuracy as 32-bit precision training on several state-of-the-art deep learning models. To demonstrate this, we compare 32-bit precision training and standard pure 16-bit training (using nearest rounding for all its operator outputs). For example, Figure 1 illustrates that for a BERT-Base model for natural language inference, the standard pure 16-bit training algorithm demonstrates 16% and 7% lower training and validation accuracies than 32-bit precision training[2]. This gap suggests that nearest rounding is the primary source of numerical error in pure 16-bit

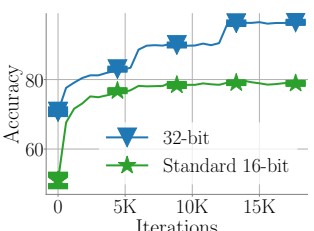

Figure 1: Standard pure 16-bit training shows lower training accuracy compared to 32-bit training on a BERT model.

algorithms, significantly degrading the convergence and model accuracy. To alleviate this problem, in Section 3, we study how nearest rounding impacts convergence, and we expose insights which lead to simple techniques to improve the model accuracy in pure 16-bit training.

---

[2]We note that 32-bit and standard 16-bit training in Figure 1 start from the same initialization with the same model accuracy. The gap at the beginning of the two curves is due to curve smoothing for visualization clarity. We refer to Appendix D.1 for the unsmoothed curves which start from the same model accuracy.

# 3 PRECISE MODEL WEIGHT UPDATES ARE ALL YOU NEED

To understand how to improve the model accuracy attained by pure 16-bit training, in this section we analyze the impact of nearest rounding, the primary source of numerical error in the standard 16-bit training algorithm. In Section 3.1, we show that when model updates are small relative to the model weights, nearest rounding for model weight updates ignores small updates and consequently impedes convergence of stochastic gradient descent. In contrast, we show that nearest rounding in the forward and backward compute can have a much weaker impact on the convergence throughout training. These insights emphasize the importance of more precise model weight updates for pure 16-bit training. To achieve such precise updates using pure 16-bit training, in Section 3.2 we consider using two simple existing numerical techniques, stochastic rounding and Kahan summation, for model weight updates. Although we use a least-squares regression model for its simplicity in exposing insights, we will show in Section 4 that our insights transfer empirically to representative deep learning models.

## 3.1 UNDERSTANDING THE IMPACT OF NEAREST ROUNDING

In this section, we use a simple least-squares regression model $\frac{1}{2n}\sum_{i=1}^{n}\|\boldsymbol{x}_i^T\boldsymbol{w} - y_i\|^2$ with batch size 1 as a proxy to expose the impact of numerical errors due to nearest rounding. First, we discuss the impact of nearest rounding for model weight updates. Then, we show that nearest rounding for forward and backward compute has comparatively much weaker influence on accuracy. More concretely, we focus on underdetermined least-squares regression problems with high dimensional model weights and input features; this is intended to reflect the overparameterized setting in modern deep learning models (Li et al., 2018; Jacot et al., 2018). In this setting, the model has the capacity to perfectly fit the dataset with $y_i = \boldsymbol{x}_i^T\boldsymbol{w}^*$ where $\boldsymbol{w}^*$ is the minimizer of $f(\boldsymbol{w})$ (**A1**). To ensure convergence with a bounded gradient variance, we assume bounded input data $\|\boldsymbol{x}_i\|^2 \leq L$ (**A2**). We let $\epsilon$ denote the machine epsilon of our floating point format, such that if $u$ and $v$ are adjacent representable numbers in our floating point format, $\epsilon|u| \leq |u - v| \leq 2\epsilon|u|$. Under this standard assumption for numerical analysis on floating point numbers (Stoer & Bulirsch, 2013), nearest rounding $\boldsymbol{Q}(\cdot)$ will have a bounded error of $|\boldsymbol{Q}(u) - u| \leq \epsilon|u|$ for any $u$ in range. To simplify the presentation, we ignore overflow and underflow in our analysis here, and disregard factors of $\epsilon^2$ (as is standard in analyses of floating point error).

**Nearest Rounding for Model Weight Updates**   When stochastic gradient descent updates model weights with nearest rounding, the model weights evolve as $\boldsymbol{w}_{t+1} = \boldsymbol{Q}\left(\boldsymbol{w}_t - \alpha\nabla f_{\sigma(t)}(\boldsymbol{w}_t)\right)$. For a weight dimension $i$, if the model update $\left[\alpha\nabla f_{\sigma(t)}(\boldsymbol{w}_t)\right]_i$ is smaller than half of the difference between $[\boldsymbol{w}_t]_i$ and its neighboring representable value in a certain precision format, nearest rounding cancels this model update. This often emerges in the mid-to-late training stage when the magnitude of gradient becomes small or learning rate decays small. Formally, we show that nearest rounding can cancel updates across all dimensions for a least-squares regression model when approaching the optimal weights in Theorem 1; we defer the proof to Appendix A.

**Theorem 1.** *Consider running one step of SGD on a least-squares regression model under assumptions **A1** and **A2**. The model weight update will be entirely canceled by nearest rounding if*

$$\|\boldsymbol{w} - \boldsymbol{w}^*\| \leq \frac{\epsilon}{\alpha L + \epsilon} \cdot \min_j \left|w_j^*\right|, \tag{1}$$

*where $w_j^*$ denotes the $j$-th dimension of the optimal solution $\boldsymbol{w}^\star = \arg\min_{\boldsymbol{w}\in\mathbb{R}^d} f(\boldsymbol{w})$. Additionally, if we run multiple steps of SGD using nearest-rounded weight updates and fixed learning rate $\alpha \leq 1/L$, then the distance of the weights $\boldsymbol{w}_t$ at any timestep $t$ from the optimum is bounded by*

$$\|\boldsymbol{w}_t - \boldsymbol{w}^*\| \geq \min\left(\frac{\epsilon(1 - \alpha L)}{\alpha L + \epsilon} \cdot \min_j \left|w_j^*\right|, \|\boldsymbol{w}_0 - \boldsymbol{w}^*\|\right).$$

Theorem 1 reveals that for the least-squares regression model, nearest rounding cancels the entire model updates when the distance towards the optimal solution $\boldsymbol{w}^*$ is small relative to the magnitude of $\boldsymbol{w}^*$. Thus in the late stage of training, the model weights can halt in a region with radius $\frac{\epsilon}{\alpha L + \epsilon}\min_j\left|w_j^*\right|$ around $\boldsymbol{w}^*$. Our lower bound shows that this region limits the convergence of SGD with nearest rounding on model weight updates for least-squares regression models. In this bound, the key property is the dependency on the step size: as the step size becomes small, this error lower bound becomes *worse*, which is the opposite of the usual effect of diminishing the step size in SGD.

Given that $\boldsymbol{w}^*$ can be arbitrarily far from the zero vector, our lower bound also reveals a substantial convergence degradation for stochastic gradient descent using floating point numbers. This specific degradation, which depends on the magnitude of $\boldsymbol{w}^*$, does not emerge in training with fixed point numbers which is a setting widely used for analyzing the impact of rounding (Hou et al., 2018; Li et al., 2017). Because the lower bound on $\|\boldsymbol{w}_0 - \boldsymbol{w}^*\|$ is also in the order of $\mathcal{O}(\epsilon)$, this bound is worse for lower precision formats with a large $\epsilon$ value. In Section 4, we will empirically show that these insights from the least-squares regression model can also generalize to deep learning models, which explains the convergence and model accuracy degradation due to the small updates cancellation in model weight updates.

**Nearest Rounding for Forward and Backward Compute** In contrast to the significant convergence degradation imposed by nearest rounding on model weight updates, we show that the nearest rounding in the gradient computation (in the forward and backward passes of backpropagation) can impact convergence minimally. To show this, we consider stochastic gradient descent with nearest rounding only for compute operations which generate activations and gradients. Here to compute the gradient for least-squares regression models, the linear layer passes the rounded activation $a_i = \boldsymbol{Q}\left(\boldsymbol{x}_i^T \boldsymbol{w} - y_i\right)$ to the loss layer. (We see no quantization error within the dot product $\boldsymbol{x}_i^T \boldsymbol{w}$ itself, as all accumulation here is done with the higher-precision accumulator of the FMAC.) In the backward stage, the loss layer feeds the rounded activation gradients $g_{a,i} = \boldsymbol{Q}(a_i)$ back to the linear layer. The weight gradient is then computed as $\nabla_{\boldsymbol{Q}} f_i(\boldsymbol{w}) := \boldsymbol{Q}(g_{a,i}\boldsymbol{x}_i) = \boldsymbol{Q}\left(\boldsymbol{Q}\left(\boldsymbol{Q}\left(\boldsymbol{x}_i^T \boldsymbol{w}_t - y_i\right)\right)\boldsymbol{x}_i\right)$. To isolate the impact of nearest rounding for activations and gradients, we do not round model weights in this setting. Formally, we show that SGD with activation and gradient rounding allows for an upper bound on $\|\boldsymbol{w}_t - \boldsymbol{w}^\star\|$ which can be substantially smaller than the lower bound in Theorem 1.

**Theorem 2.** *Consider running multiple steps of SGD on a least-squares regression model under assumptions A1 and A2, using nearest rounding for only forward and backward compute, but exact arithmetic for model weight updates. Then if the step size is small enough that $\alpha \leq 1/L$, the distance of the weights $\boldsymbol{w}_t$ at any timestep $t$ from the optimum will be bounded by*

$$\mathbf{E}\left[\|\boldsymbol{w}_t - \boldsymbol{w}^*\|^2\right] \leq \exp\left(-\alpha\mu t\left(1 - \tfrac{4\epsilon L}{\mu}\right)\right) \cdot \|\boldsymbol{w}_0 - \boldsymbol{w}^*\|^2,$$

*where $\mu$ is the smallest eigenvalue of the data covariance matrix $\frac{1}{n}\sum_{i=1}^n \boldsymbol{x}_i \boldsymbol{x}_i^T$.*

As $t$ can be made arbitrarily large, this bound guarantees us substantially more accurate solutions than the lower bound attained by using nearest rounding only for model weight updates in Theorem 1. This *shows that rounding for the weight updates is the primary source of error*, as even with all other operations quantized, the algorithm is guaranteed to converge closer to the optimum than is even possible with just the weight updates rounded with nearest rounding. Note that the bound in Theorem 2 is able to guarantee arbitrarily accurate solutions because we ignore underflow here. In practice, precision would eventually be limited by underflow even in the setting of Theorem 2; however, the underflow threshold for BFloat16 is small enough that this represents a level of error that deep learning applications are generally able to tolerate. We refer to Appendix A for the proof.

**Theory Validation** To validate our insights, we compare the impact of nearest rounding for model weight updates against that of nearest rounding in forward and backward compute on a synthetic 10-dimensional least-squares regression problem. Specifically, the input data are sampled from a zero-mean unit-variance normal distribution while the model weight is generated from a uniform distribution in the range of $[0, 100)$. We perturb the label with a zero-mean normal distribution with standard deviation $0.5$. As shown in Figure 2, when using a learning rate $0.01$ and 16-bit nearest rounding for model weight updates, we observe that the training loss saturates at a magnitudes higher level than stochastic gradient descent without rounding because of updates cancellation. Meanwhile, when using nearest rounding only for forward and backward compute, the loss saturates at a level close to that attained by training without rounding. These observations align with our insights on the relative impact of nearest rounding for model weight updates and for forward and backward compute.

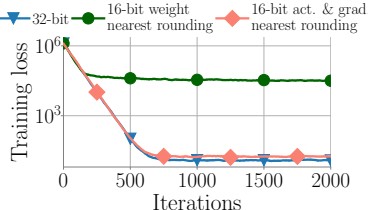

Figure 2: **Theory validation**. On a least square regression model, (smoothed) training losses with 16-bit nearest rounding for weight updates saturate at a higher level than 32-bit training. With only using nearest rounding for forward and backward compute, the losses saturate much closer to 32-bit training.

## 3.2 High-accuracy Pure 16-bit Training

In Section 3.1, we showed that nearest rounding for model weight updates is the bottleneck for convergence in the standard pure 16-bit training algorithm; this is because it cancels small model updates which degrades the model weight precision. This motivates us to consider two existing techniques, *stochastic rounding and Kahan summation* (Kahan, 1965) for improving weight updates. These techniques have been reliably applied in different numerical domains (Hopkins et al., 2020; Antonana et al., 2017) and can hypothetically enable high-accuracy pure 16-bit training. We present details on how to integrate these techniques into SGD and AdamW optimizers with pure 16-bit model weights and optimizer states such as momentum in Appendix B.

**Stochastic Rounding** Stochastic rounding for floating point numbers has been used in training certain model components (Zhang et al., 2018) and can potentially improve the model accuracy of pure 16-bit training for general models. Specifically, let $\mathbb{S}$ be the set of all values representable by a limited precision format: the upper and lower neighboring values for $a \in \mathbb{R}$ are $a_u = \min_{x \geq a, x \in \mathbb{S}} x$ and $a_l = \max_{x \leq a, x \in \mathbb{S}} x$. Stochastic rounding randomly rounds $a$ up to $a_u$ with probability $(a - a_l)/(a_u - a_l)$ and otherwise rounds down to $a_l$. We consider pure 16-bit training using stochastic rounding only for the subtraction output in the model update $\boldsymbol{w}_t - \alpha \nabla f_{\sigma(i)}(\boldsymbol{w}_t)$. We keep nearest rounding for all the other compute operations. Here, the rounded model weight is an unbiased estimate of the precise value, so it will still make progress in expectation; this prevents the halting effect from nearest rounding on model weight updates. We note that in modern hardware, stochastic rounding can be implemented without any expensive multiplication or division arithmetic (De Sa et al., 2017). Thus using stochastic rounding for model weight updates adds minimal overhead when training modern deep learning models; we discuss explicitly how to achieve this in Appendix B.1.

**Kahan Summation** The Kahan summation algorithm uses an auxiliary variable to track numerical errors and to compensate the accumulation results. In the context of pure 16-bit training, we use a 16-bit auxiliary variable $\boldsymbol{c}_t \in \mathbb{R}^d$ to track the error in model weights. To ensure pure 16-bit data paths, we keep nearest rounding for all operators in compute graphs, including those during Kahan accumulation in Algorithm 1. At iteration $t$, we first compensate the current model

---

**Algorithm 1** SGD updates with Kahan summation

1: Auxiliary value $\boldsymbol{c}_0 \leftarrow 0$ at initialization
2: **Input:** Model updates $-\alpha \nabla f_{\sigma(i)}(\boldsymbol{w}_t)$ at iter. t
3: $\boldsymbol{u}_{t+1} \leftarrow -\alpha \nabla f_{\sigma(i)}(\boldsymbol{w}_t)$
4: $\boldsymbol{y}_{t+1} \leftarrow \boldsymbol{u}_{t+1} - \boldsymbol{c}_t$ ▷ Compensate updates
5: $\boldsymbol{s}_{t+1} \leftarrow \boldsymbol{w}_t + \boldsymbol{y}_{t+1}$ ▷ Accumulate updates
6: $\boldsymbol{c}_{t+1} \leftarrow (\boldsymbol{s}_{t+1} - \boldsymbol{w}_t) - \boldsymbol{y}_{t+1}$ ▷ Measure errors
7: $\boldsymbol{w}_{t+1} \leftarrow \boldsymbol{s}_{t+1}$
8: **Return:** $\boldsymbol{w}_{t+1}$

---

update $\boldsymbol{u}_{t+1}$ by subtracting the previous error $\boldsymbol{c}_t$. We then compute the new model weights by adding the compensated updates $\boldsymbol{y}_{t+1}$ to the current weights $\boldsymbol{w}_t$. We reversely subtract previous model weights $\boldsymbol{w}_t$ and the compensated updates $\boldsymbol{y}_{t+1}$ to acquire the new numerical error $\boldsymbol{c}_{t+1}$ in the updated weights $\boldsymbol{w}_{t+1}$. For small updates $\boldsymbol{u}_t$ which would cause no change in the weights after nearest rounding, this reverse subtraction records the canceled updates in the error $\boldsymbol{c}_{t+1}$. Across iterations, small updates can be accumulated in $\boldsymbol{c}_{t+1}$ until $\boldsymbol{c}_{t+1}$ grow large enough to affect the model weights; this allows convergence to continue when it would otherwise halt due to nearest-rounding effects. In spite of the additional auxiliary value, pure 16-bit training with Kahan summation for model weight updates can still have advantages in terms of throughput and memory consumption compared to 32-bit and mixed precision training; we refer to Appendix B.2 for details.

## 4 Experiments in Deep Learning

Our theory in Section 3 reveals that nearest rounding on model weight updates is the primary source of numerical error during training. This motivates us to suggest using stochastic rounding and Kahan summation in pure 16-bit training for improved model accuracy. To first validate our theory, in this section we start by demonstrating that by ablating nearest rounding on model weight updates from the standard 16-bit training algorithm, the model accuracy gap compared to 32-bit precision training can be closed on deep learning models. Next, we show empirically that with either stochastic rounding or Kahan summation on model weight updates, pure 16-bit training can match the accuracy of 32-bit precision training across representative deep learning applications.

**Experiment Setup** To validate the accuracy bottleneck, we consider three representative models: ResNet-18 (He et al., 2016) on the CIFAR10 image classification (Krizhevsky et al., 2009), BERT-

Table 1: **Model accuracy bottleneck for the standard pure 16-bit training algorithm.** This algorithm shows validation accuracy gap compared to 32-bit training. In an ablation of this algorithm, we use 32-bit model weights and turn off nearest rounding only on model weight updates. This eliminates the gap, suggesting that nearest rounding on model weight updates is the accuracy bottleneck.

| Model | Dateset (Metric) | 32-bit | Standard 16-bit | Standard 16-bit & 32-bit weights |
|---|---|---|---|---|
| ResNet-18 | CIFAR10 (Acc%) | $95.45 \pm 0.07$ | $94.23 \pm 0.12$ | $95.40 \pm 0.05$ |
| DLRM | Kaggle (AUC%) | $80.27 \pm 0.01$ | $78.49 \pm 0.08$ | $80.26 \pm 0.01$ |
| BERT-Base | MNLI (Acc%) | $84.26 \pm 0.08$ | $77.53 \pm 0.07$ | $84.34 \pm 0.04$ |

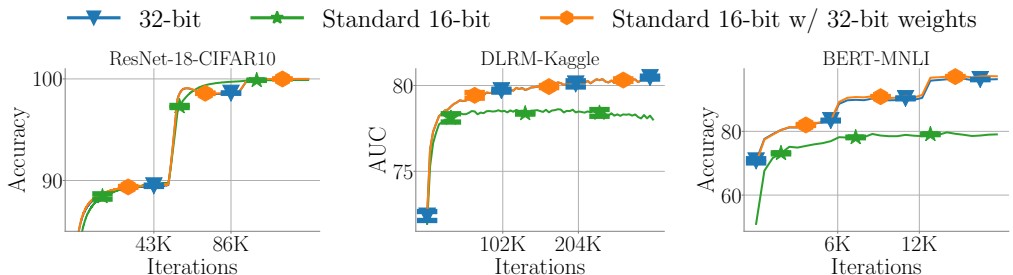

Figure 3: **Training accuracy gap imposed by the standard pure 16-bit training.** The standard algorithm fails to match the training accuracy of 32-bit training, especially in the middle-to-late stage. We close this accuracy gap by ablating nearest rounding for weight updates from the standard algorithm. This indicates that nearest rounding for model weight update is the accuracy bottleneck.

Base (Devlin et al., 2018) on the MNLI natural language inference (Wang et al., 2018a), and DLRM model (Naumov et al., 2019) on the Kaggle Advertising Challenge (CriteoLabs, 2014). To extensively evaluate pure 16-bit training with stochastic rounding and Kahan summation, we additionally consider larger datasets and more applications: ResNet-50 on the ImageNet (Deng et al., 2009), BERT-Base on the Wiki103 language model[3] (Merity et al., 2016), DLRM model on the Criteo Terabyte dataset (CriteoLabs, 2018), and Deepspeech2 (Amodei et al., 2016) on the LibriSpeech datasets (Panayotov et al., 2015). As there is no publicly available accelerator with the software and hardware support necessary for our study, we simulate pure 16-bit training using the QPyTorch simulator (Zhang et al., 2019). QPyTorch models PyTorch kernels such as matrix multiplication as compute graph operators, and effectively simulates FMAC units with 32-bit accumulators[4]. For all training algorithms, we use the same hyperparameters as their original papers or code repositories. We report statistically meaningful results with averaged metrics and standard deviations across runs with 3 random seeds. We refer to Appendices C and D for experiment details and extended results.

**The Model Accuracy Bottleneck** To validate our insights from Section 3, we first show empirically that nearest rounding on the model weights is the primary model accuracy bottleneck on several deep learning models. To do this, we keep the model weights in 32-bit precision and turn off nearest rounding on the model weight updates while keeping nearest rounding for all other operators in the compute graph. Figure 3 shows that the standard pure 16-bit training algorithm with nearest rounding on all operators has up to 16% training accuracy gap compared to 32-bit training. Although this gap can be small in the early training phase, it grows larger in later stages. In contrast, by ablating nearest rounding on model weight updates, the standard algorithm can fully match the training accuracy attained by 32-bit training. We notice in Table 1 that this ablation can also close the 1.2% to 6.7% validation accuracy gap when comparing the standard pure 16-bit training to 32-bit training. These observations validate our insights from Section 3.1 and motivate the use of stochastic rounding and Kahan summation on model weight updates.

---

[3]We subsample 25% of the Wiki103 and 100 hours of Librispeech training set because of the training time.
[4]Following the convention in mixed precision training (Micikevicius et al., 2017), our simulator uses fused operators for computationally inexpensive activation and normalization layers.

Table 2: **Pure 16-bit training can match 32-bit training on model accuracy.** With stochastic rounding or Kahan summation for model weight updates, pure 16-bit training can attain $0.1\%$ lower to $0.2\%$ higher absolute value for validation accuracy metrics across applications.

| Model | Dateset (Metric) | 32-bit | 16-bit Stochastic | 16-bit Kahan | Standard 16-bit |
|---|---|---|---|---|---|
| ResNet-18 | CIFAR10 (Acc%) | $95.45 \pm 0.07$ | $95.33 \pm 0.08$ | $95.36 \pm 0.07$ | $94.23 \pm 0.12$ |
| ResNet-50 | ImageNet (Acc%) | $75.70 \pm 0.05$ | $75.45 \pm 0.03$ | $75.61 \pm 0.14$ | $67.10 \pm 0.24$ |
| DLRM | Kaggle (AUC%) | $80.27 \pm 0.01$ | $80.18 \pm 0.02$ | $80.26 \pm 0.01$ | $78.49 \pm 0.08$ |
| | Terabyte (AUC%) | $80.32 \pm 0.00$ | $80.25 \pm 0.00$ | $80.32 \pm 0.00$ | $78.79 \pm 0.02$ |
| BERT | MNLI (Acc%) | $84.26 \pm 0.08$ | $84.35 \pm 0.12$ | $84.45 \pm 0.03$ | $77.53 \pm 0.07$ |
| | Wiki103 (PPL) | $5.50 \pm 0.50$ | $5.84 \pm 0.53$ | $5.45 \pm 0.51$ | $56.88 \pm 1.77$ |
| DeepSpeech2 | Librispeech (WER) | $62.71 \pm 0.07$ | $62.85 \pm 0.07$ | $62.87 \pm 0.18$ | $69.42 \pm 0.22$ |

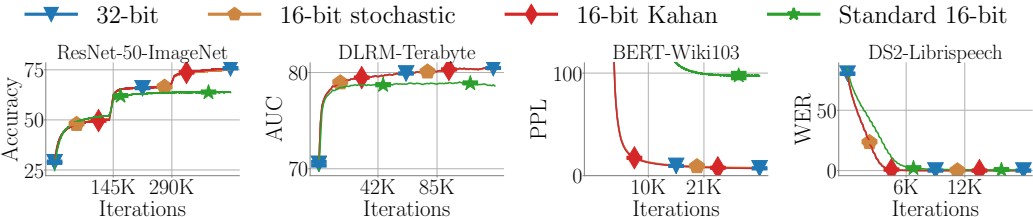

Figure 5: **Training accuracy for pure 16-bit training.** With stochastic rounding or Kahan summation enabled for model weight updates, pure 16-bit training matches 32-bit precision training in terms of training accuracy with negligible differences across the applications in our experiments.

**High-accuracy Pure 16-bit Training**   Next, we validate empirically that enabling stochastic rounding or Kahan summation for model weight updates allows pure 16-bit training to attain matching model accuracy as 32-bit training. In Table 2, we first show that by using stochastic rounding for model weight updates, pure 16-bit training matches the validation accuracy of 32-bit training with at most $0.1\%$ difference on the CIFAR10, Kaggle, Terabyte, MNLI and Librispeech datasets, a majority of the applications in our experiments. For applications where stochastic rounding still shows a non-negiglable accuracy gap with more than $0.1\%$ discrepancy, we show that Kahan summation for model weight updates can enable pure 16-bit training to match the model accuracy of 32-bit training algorithms. We show that Kahan summation for model weight updates can boost pure 16-bit training to higher validation accuracy than using stochastic rounding. More concretely, the Kahan summation for model weight updates shows $0.2\%$ higher top-1 accuracy and $0.1\%$ higher AUC respectively for ResNet-50 on ImageNet and for recommendation on Terabyte than using stochastic rounding. Consequently as shown in Table 2, by using Kahan summation for model weight updates, pure 16-bit training match the model accuracy attained by 32-bit precision training across all the applications in our experiments. This validates that stochastic rounding and Kahan summation can enable pure 16-bit training algorithms to match the model accuracy of 32-bit training.

**Memory efficiency and model accuracy trade-off**   Additionally, we show that stochastic rounding and Kahan summation can be combined for pure 16-bit training, which exposes a memory efficiency and model accuracy trade-off for practitioners to exploit. As an example, in Figure 4 we demonstrate this trade-off by incrementally replacing stochastic rounding with Kahan summation on various model weights in the DLRM model on the Kaggle dataset. As we apply Kahan summation to more model weights, the weight memory cost increases by up to $2\times$. As this cost increases we also observe up to $0.04\%$ improvement in AUC. This exploits a memory efficiency and model accuracy trade-off that users should consider when deciding which technique to leverage.

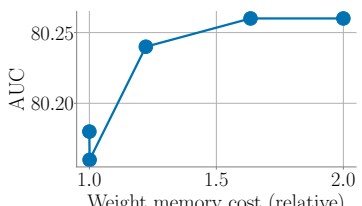

Figure 4: **Efficiency and accuracy trade-off.** With stochastic rounding and Kahan summation on different parts of the DLRM-Kaggle, it attains higher model accuracy at the cost of more weight memory.

## 5 RELATED WORK

There is a plethora of research work on low-precision training for deep learning models. On certain specific models such as convolutional or recurrent neural networks, training with fixed point or floating point precisions lower than 16-bit has been shown to be feasible when using customized techniques (Wang et al., 2018b; Zhou et al., 2016; Hubara et al., 2017; Courbariaux et al., 2014; Ott et al., 2016; Sun et al., 2019). Instead of proposing new techniques for specific model types using lower than 16-bit precision, we focus on finding the minimal set of simple techniques required by generic mode training on future general-purpose deep learning accelerators requiring only modern 16-bit compute units. Such emerging accelerators have the potential to unlock substantially improved hardware efficiency compare to those still requiring 32-bit compute units.

Recent work shows that in low precision training, stochastic gradient descent with stochastic rounding for model weight updates only degrades worst-case upper bounds on convergence minimally; this shows stochastic rounding is an effective technique to attain strong model accuracy in low precision training (Li et al., 2017; Hou et al., 2018). Our analysis is orthogonal and complementary to these upper bounds. Specifically, we prove a lower bound on convergence when using standard nearest rounding for model weight updates. This lower bound shows that nearest rounding for weight updates can substantially degrade the convergence even in the most optimistic case no matter how learning rates are tuned. We use this lower bound together with the previous upper bounds to inform future accelerator designers that only supporting nearest rounding is not enough to maximize model accuracy; to alleviate this problem, stochastic rounding for model weight updates is one of the minimal supports required in future training accelerators.

## 6 CONCLUSION

In this paper we study pure 16-bit training algorithms that require only 16-bit compute units. We show that nearest rounding on model weight updates is the primary cause of convergence and model accuracy degradation in standard pure 16-bit training. To alleviate this issue, we apply two existing techniques: stochastic rounding and Kahan summation. With these techniques, we demonstrate that pure 16-bit training can match the model accuracy of 32-bit precision training across many deep learning models. Our study suggests that it is feasible to design high-accuracy deep learning accelerators using only 16-bit compute units if stochastic rounding and Kahan algorithm are supported.

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
