# OpenReview forum: "Revisiting BFfloat16 Training"
_ICLR.cc/2021/Conference — Reject_

### Official Review · AnonReviewer3 · 2020-10-26
**This submission rehashes BF16 labeling it incorrectly as 'pure 16b' and creating the wrong perception that they are first to question the so-called 'folklore' of 16b MAC not enough for training.**

**Rating:** 3
**Confidence:** 5

**Review:**

During the rebuttal, I concluded that this submission is highly confusing, rather misleading.  I was led to believe the authors are in fact talking 'pure 16b MAC' - meaning 16b FP multiplies and 16b accumulate. After reading their responses to R4, I now learnt that they in fact are using 32b accumulates as is already standard practice in BF16 - If so, they have little or nothing new to offer. Rounding discussions the paper focuses on become highly secondary.  BF16 is already well-understood and accepted. Their writeup was highly misleading to say the least. I change my rating based on this.

They use a representative set of benchmarks which include, CNN-based Resnet, recommendation proxy DLRM, and NLP proxy BERT.

Novelty is limited, but critical: They reiterate prior observations that accurate weight updates are more critical for maintaining accuracy than forward-backward gradient updates. Former is what suffers when round-to-nearest cancels out small updates. Stochastic rounding has also been published before and shown to still miss the accuracy mark with >0.,1% accuracy gap for some important benchmarks. Novel part of this work is suggesting a Kahan summation based software fix on top of stochastic rounding to overcome the accuracy loss.

Primary issue: This is a very confusing and misleading writeup.  Authors should have clearly spelt out that they are in fact talking about BF16 - which is well-publicized for years now as 16b mule and 32b accumulate - and not labelled it as 'pure 16b MAC' - which is already known to exist as well, and proven to be not sufficient for DL training.  In light of this, this work has very little value-add. I change my rating now to 'clear reject' for their misleading writing style.

---

> ### Author Response · Authors · 2020-11-18
> **Response to Reviewer 3**
>
> We thank R3 for the favorable consideration on the importance of insights from our study. We refer to the general reply for questions shared across reviewers and resolve the remaining comments in the below.
>
> We agree with R3 that FP32 support is needed for general purpose processors. However, we believe eliminating the requirement of 32-bit compute units can be of practical importance for accelerators specialized for training generic deep learning models. E.g. as discussed in the general reply for common questions, both 32-bit training and mixed precision training require compute units with 32-bit multiply for operations inside optimizers. In contrast, the 16-bit training algorithms we study only require faster and more energy efficient units with 16-bit multiply; this benefit is especially pronounced for training large SOTA models using optimizers with sophisticated multiplications such Adam.  As accelerators for DL training are active research topics and open a large industrial market for both startups and hardware giants, we are very excited to continuously explore the feasibility of accelerators only using <32 bit compute units. To clarify the above, we will add discussions in the preliminary section.

---

### Official Review · AnonReviewer1 · 2020-10-27
**Limited novelty, but extensive empirical evaluation**

**Rating:** 6
**Confidence:** 3

**Review:**

### Summary
This work reinvigorates half precision training as an alternative to either full single precision or mixed half and single precision. The authors demonstrate that the nearest rounding is the culprit for the worse performance of half precision training compared to single precision, due to cancelling small updates. They then propose two known techniques that can mitigate this effect, stochastic rounding and Kahan summation. Empirically, they demonstrate on an extensive suite of tasks that the proposed adjustments lead to half precision performance that is almost on par to single precision.

### Pros
- Solid experimental evaluation and results on a large variety of tasks
- Both modifications that need to done are simple and straightforward

### Cons
- Limited novelty

### Recommendation
Overall, even though the paper does not have a lot of novel content, the experimental evaluation is thorough and demonstrates that the improvements close the gap to single precision training. Therefore I am keen on accepting this work, although admittedly not by much.

### Detailed feedback
The paper is relatively well written and easy to follow. The authors nicely motivate their modifications which clearly show that the gap between half and single precision training is almost closed. My main point of criticism is that the novelty of this paper is relatively small, as the techniques proposed for performing the quantized weight update are, firstly, not new and, secondly, have been used in previous works for similar reasons. More specifically, performing stochastic rounding for the weight update in order to make progress when the magnitude of the update is small (thus rounded to zero) was also proposed at [1], which also showed successful 16 bit training (albeit for outdated tasks). As for Kahan summation; that has been proposed before for training quantised neural networks at [2] (again as a means to avoid making no progress when the parameter update is small). As a result, the true contributions of this work lie on the extensive empirical evaluation along with the theoretical analysis of rounding in a simple linear model.

As for other feedback and questions
- At section 3.1 you first assumption A1 seems peculiar, as you state that you work on a least squares regression problem, but you assume that the model is overparametrized. How can this be the case? In a linear model, the amount of parameters is bounded by the dimensionality of the input (and it doesn’t seem that you perform any feature expansion for x). I believe that a more reasonable statement would be to assume that the actual data are generated by a linear model, hence there exists a true solution w^*.
- Stochastic rounding requires the generation of random numbers; can this step also be done accurately in bfloat16?
- From the evaluation it seems that Kahan summation performs better but it increases the memory size by a factor of 2; how does this fare memory wise to having the weights in single precision and is it a worthy tradeoff?
- I believe that comparisons against other methods for quantised training that use fixed point instead of floating point would be interesting, e.g., [3, 4]. This would highlight the differences between these two formats and show whether the more expensive floating point operations are necessary.

[1] Deep Learning with Limited Numerical Precision, S. Gupta, A. Agrawal, K. Gopalakrishnan, P. Narayanan, 2015
[2] Training Deep Neural Networks in Limited Precision, H. Park, J. H. Lee, Y. Oh, S. Ha, S. Lee, 2018
[3] Training and Inference with Integers in Deep Neural Networks, S. Wu, G. Li, F. Chen, L. She, 2018
[4] Per-Tensor Fixed-Point Quantization of the Backpropagation Algorithm, C. Sakr, N. Shanbhag, 2018

---

> ### Author Response · Authors · 2020-11-18
> **Response to Reviewer 1**
>
> We thank R1 for the thoughtful comments and refer to the general reply for shared questions; we resolve the remaining comments in the below.
>
> R1 raises the question when a least square regression will be overparameterized and suggests alternative elaboration on the overparameterized assumption. Our overparameterized assumption refers to the underdetermined least-squares regression models. In such cases, the model dimensionality (input feature dimension) is fixed but can be a very large value (relative to the number of samples). We consider this setting to reflect the overparameterized nature of deep neural networks. We agree with R1 that alternatively it is also very intuitive to elaborate the same technical condition as an assumption where the training data is generated from an underlying linear model. We will better elaborate on the overparameterization assumption in the beginning paragraph in Section 3.1.
>
> We agree with R1 that it is intrinsically interesting to compare floating point low precision training algorithms to fixed point ones. Though the focus of our paper is on floating point training algorithms which are widely adopted in emerging model-agnostic accelerators, we are very eager to explore this comparison in future work to reveal more insights for accelerator designers in an even further horizon.

---

> > ### Comment · AnonReviewer1 · 2020-11-23
> > **Response to rebuttal**
> >
> > Thank you for clarifying and responding to my feedback. I believe the picture is a bit clearer now, but the revised draft is still missing so I cannot see the updates. Overall, my view of this submission has not changed significantly, so I will not raise my score further.

---

> > > ### Author Response · Authors · 2020-11-24
> > > **Revised draft updated.**
> > >
> > > Thank you for the prompt reply and helpful discussion. We just uploaded the revised draft with comments incorporated for your further review and reference.

---

### Official Review · AnonReviewer2 · 2020-10-28
**A good submission but lack novelty**

**Rating:** 5
**Confidence:** 5

**Review:**

This paper explores the possibilities of reducing the precision of the weight update operation (i.e. AXPY ops) from 32 bit to 16 bit in today’s BFloat16 training framework. To enable 16 bit update, the authors proposed two techniques, i.e. stochastic rounding and Kahan summation. The authors use a simple least-squares regression model to theoretically explain the model accuracy degradation due to nearest rounding, then experimentally demonstrate the effectiveness of two techniques on a range of deep learning models and dataset.

Strong points:

This paper is very well written. The problem is well addressed, and the solutions are well explained. The authors also provided both theoretical analysis and comprehensive experimental results.


Week points:

1). Novelty: Both the problem addressed, and the solutions proposed in this paper have been reported in recent publications. In particular, both (https://papers.nips.cc/paper/7994-training-deep-neural-networks-with-8-bit-floating-point-numbers) and (https://papers.nips.cc/paper/8736-hybrid-8-bit-floating-point-hfp8-training-and-inference-for-deep-neural-networks) investigated the reduced precision of weight update from 32 bit to 16 bit or less. Moreover, the former introduced stochastic rounding and the later introduced round-off residue which basically the same as the Kahan accumulation technique discussed in this paper. Although, both papers discussed this topic in the FP8 training frameworks, while this paper in the BFloat16 framework, the basic concepts are the same. The authors of this paper summarized the techniques nicely, however, the novelty limited.

2). On the same note, theoretical analysis on the impact of rounding mode on quantized weight update were also discussed in recent publications, such as (https://arxiv.org/abs/1706.02379 and https://openreview.net/forum?id=ryM_IoAqYX). It would have been nice to include these discussions as background knowledge and to distinguish this work from others.

To clarify:

1). The proposed Kahan Summation method created a second tensor to store/accumulate the quantization error. Both weight and rounding error tensors are in 16 bit which in total, effectively, is 32 bit. Since AXPY is a very fast operation, this method does not seem to save much in terms of memory or speed.

2). Today, SGD is often used with momentum, could the authors comment on the precision of momentum accumulation. And how about other popular optimizers, such as Adam?

3). The analysis, i.e. Theroem 1 and 2, is based on a simple least-square regression model. Can this theory generalize to deep learning models?

4).  On Table 2, last two rows of last column, the value seems to be inconsistent with other data in the same row.

---

> ### Author Response · Authors · 2020-11-18
> **Response to Reviewer 2**
>
> We thank R2 for the detailed feedback on both the theory and empirical aspects. We refer to the general reply for questions in common and discuss the remaining questions in the below.
>
> R2 asked what is the precision of optimizer states, such as momentum in SGD. In our experiments, we also use BFloat16 precision for momentum in the SGD optimizer, and for first / second moments in the Adam optimizer. This ensures that all the optimizer operations can directly use 16-bit input and generate 16-bit output using 16-bit compute units. To make this more clear, we will increment the discussion on the optimizer implementation using stochastic rounding or Kahan summation for model weight updates in Appendix B.
>
> Regarding the generalization of our analysis on least-squares models, the insights that nearest rounding for model weight updates degrades convergence due to cancellation of updates empirically generalize to deep learning models. E.g. in Appendix D.3, we show that up to 80% of model updates can get cancelled due to BFloat16 nearest rounding for weight updates in a DLRM recommender model; this leads to ~3% AUC degradation compared to 32-bit training. We are also excited to study rigorous theory generalization to deep learning models feasible for analysis as future works.
>
> R4 also very sharply noticed that in Table 2, standard 16-bit training demonstrates one magnitude worse validation perplexity on Wiki103 datasets than 32-bit training. This large discrepancy is valid. It is because that validation perplexity value is exponential with respect to the validation loss. Thus the difference of validation loss can be significantly magnified by validation perplexity which is the standard metric for language modeling tasks.

---

### Official Review · AnonReviewer4 · 2020-11-02
**Serious problems with the approach**

**Rating:** 3
**Confidence:** 4

**Review:**

I think the use of QPyTorch for the experiments here invalidates the results since the intermediate matrix multiplies are done in single precision (FP32), and so are more optimistic than a pure 16-bit implementation. (This is both according to the authors Sec 4, experiment setup; and according to the QPyTorch paper arxiv:1910.04540, Sec 3 intro.) For these kinds of experiments to be meaningful, they have to be done on native 16-bit hardware which luckily is becoming more common, e.g., Google's TPUs or the newer NVIDIA GPUs.

There are two other problems. First, it is not clear how stochastic rounding would be implemented in hardware. Doing it for every MAC operation could likely be even more expensive than just doing 32-bit MAC operations, since it involves the generation of random numbers, division, etc. Second, Kahan summation takes up twice the weight storage, so a more detailed calculation is needed to compare any hardware/energy savings to use that instead of just 32-bit.

As an aside, it may be interesting in Figure 1 to zoom in on the initial part of training to understand where the difference between 32-bit and standard 16-bit comes from in early training since at that point, the gradients are generally larger than later on in training.

---

> ### Author Response · Authors · 2020-11-18
> **Response to Reviewer 4**
>
> We thank R4 for the detailed feedback and discussion. We refer to the general reply on common questions and resolve the other comments in the following.
>
> R4 suggested that we should run experiments with TPUs or recent GPUs to achieve evaluation when the accumulation operation in MAC is also in 16-bit precision. Unfortunately these hardware accelerators (like all of the existing ones that we know of) use higher precision accumulators in MAC units. In more details, the BFloat16 unit we considered takes 16-bit input for multiplication and uses 32-bit higher precision for the accumulation in MAC operations.
>
> Such higher precision accumulation is inexpensive compared to multiplication but important to the numerical accuracy of operations such as matrix multiplication and convolutions. Thus it is the standard design practice in today’s TPUs and GPUs [6, 8]. Because there is minimal reward from a hardware design perspective in eliminating higher precision accumulators, studying 16-bit precision accumulation in MAC operation is beyond the scope of this paper (albeit still very interesting conceptually). Therefore, QPyTorch can support exact compute simulation for the BFloat16 units [9] in our study that exist (and will continue to exist) in modern hardware; we use it to flexibly evaluate numerical techniques. Using TPUs or new GPUs would still result in the same numerical output for BFloat16 units as in the QPyTorch because these accelerators are also using 32-bit instead of 16-bit MAC accumulation [6, 8]. We thank R4 for bringing up this point as we very much agree that it needs to be addressed more clearly and explicitly in our draft and we will modify the introduction to address this point.
>
> R4 also noticed that the training accuracies are different at the early stage for 32-bit and standard 16-bit training in Figure 1. This is because we use strong smoothing for the curves for clear visualization. Both training algorithms start from the same initialization and the training accuracy gradually diverges. We will include a less smoothed curve in the appendix D.1 to clarify this observation and link from the main paper.

---

> > ### Comment · AnonReviewer4 · 2020-11-24
> > **Not sure about that**
> >
> > Thank you for the response. I do not understand this statement in the paper:
> >
> > > Importantly, for a 16-bit FMAC unit, the accumulator a has higher-than-16-bit
> > precision. This higher precision accumulator is standard in modern hardwares because it ensures
> > the precision of complex operators like matrix multiplication (Henry et al., 2019; Markidis et al.,
> > 2018). The result in the accumulator then needs to be rounded to 16-bits before it is output from
> > the FMAC unit (e.g. before writing to memory).
> >
> > Looking at the documentation on NVIDIA's site it seems certainly possible to do the MAC in 16 bits (as one might naively expect: [NVIDIA reference for bfloat fma](https://docs.nvidia.com/cuda/cuda-math-api/group__CUDA__MATH____BFLOAT16__ARITHMETIC.html#group__CUDA__MATH____BFLOAT16__ARITHMETIC_1g463bf603ed3b2eba19de9ab7d37aad44)
> >
> > ---
> >
> > Now, it is a different argument to say that you are targeting mixed precision training where the multiplies happen in BFloat16 and the accumulates in 32 bits, but then I think the title of the paper should be "Revisiting Mixed Precision Training with BFloat16 and FP32 accumulates" or something like that, and I don't think you can say "pure 16-bit training" as you do through out the paper.
> >
> > But in that case the main point of comparison should be other mixed-precision training (and not 32-bit training) such as perhaps the following:
> >
> > [Mixed-Precision Training of Deep Neural Networks](https://developer.nvidia.com/blog/mixed-precision-training-deep-neural-networks/)
> >
> > Note that they say on that page:
> >
> > > We found that accumulation into single precision is critical to achieving good training results. Accumulated values are converted to half precision before writing to memory.
> >
> > Thus I think it is important to NOT be doing accumulates in FP32 if one wants to claim 16-bit purity (since this mixed mode _is_ what makes it mixed-precision).
> >
> > And perhaps it is possible that your technique may help the NVIDIA method above by not requiring the FP32 shadow copy of weights.

---

> > > ### Author Response · Authors · 2020-11-24
> > > **To the best of our knowledge our statement is correct**
> > >
> > > We thank the reviewer for the prompt reply. The linked does show input and output types but the description of what happens in the hardware for each instruction is still opaque. To support our statement, we refer to more details here:
> > > https://developer.download.nvidia.com/video/gputechconf/gtc/2020/presentations/s21730-inside-the-nvidia-ampere-architecture.pdf
> > >
> > > The document above (slides 9-15) goes through the accumulator widths that we are referencing (specifically for GPUs). For training, this official Nvidia document shows that they use a FP32 accumulator for the tensor cores in both A100 and V100 **for all training types**. There was one FP16 accumulator added **only for inference** in the new A100, but there are no added TOPs in the table (slides 12). Regardless, our original claim of FP32 (higher precision) accumulators being the de-facto hardware support for training holds true.
> > >
> > > It is important to note that the pure 16-bit training we study is not the same as the mixed precision literature linked above. The details above are hardware support, hidden from users and default in all ML accelerators that we are aware of. The mixed precision work introduces FP32 model weights and optimizer states that need to be stored in memory (increasing memory units and requiring real hardware FP32 unit support, not just a higher-precision accumulator). This is why our work, in comparison, is named as pure Bfloat16 training as discussed in the intro and preliminary sections.

---

### Author Response · Authors · 2020-11-18
**General response to shared comments (1/2)**

We thank the reviewers for their thoughtful comments and detailed questions. First, we address questions and comments common among multiple reviewers in this reply. Next, we discuss the individual comments for each reviewer one by one.

In this paper, we study whether 16-bit training, which requires only 16-bit compute units for emerging deep learning accelerators, can match the model accuracy attained by 32-bit training. Such accelerators promise better efficiency (e.g. chip area, energy consumption) or more overall available compute capacity than accelerators which still require 32-bit floating point compute support.

**The goal of our paper** is not to propose new techniques to enable < 16 bit precision training for certain types of models [1, 2]. Instead, we aim to reveal what is the minimal (and simplest) set of model-agnostic algorithmic techniques to achieve strong accuracy for SOTA models on emerging/future accelerators, robustly across many model domains (e.g. CV, NLP, Speech, Recommender). Towards this end, our study challenges the practitioner’s folklore that mixed precision (16-bit & 32-bit) is needed to maximize model accuracy for training generic DL models. This suggests the feasibility of new deep learning accelerators that only require modern 16-bit units for generic model training.

It is important to note that these widely-adopted modern 16-bit units do not eliminate 32-bit higher precision accumulation in MAC operations. This higher precision accumulation is inexpensive compared to 16-bit multiply with respect to hardware design in modern 16-bit MAC units but is critical to the numerical accuracy of operations such as matrix multiplication and convolution. As a result, this higher precision accumulation in MAC units will likely continue to be standard in all modern hardware accelerators (including GPUs and TPUs) [6, 7, 8]. To the best of the authors knowledge, we are the first to provide such a study to inform hardware designers about the minimal algorithmic support needed for model-agnostic accelerators that can train SOTA models across application domains using only 16-bit compute units. We agree with the reviewers that these points should be made explicitly and clearly in our draft and we will modify the introduction of our draft accordingly.

**The goal of our theoretical analysis** is different and complementary to the previous theory in [3, 4]. The theories in [3, 4] prove upper bounds on convergence. These upper bounds show that stochastic rounding for weight updates can achieve good convergence even in the worst-case. On the other hand, we prove lower bound on convergence for standard nearest rounding for weight updates. This lower bound shows that nearest rounding for weight updates can substantially degrade the convergence even in the most optimistic case. By combining the insights from our lower bound with those from previous upper bounds, we can finally close the loop and inform the future accelerator designers that 1) only supporting nearest rounding is not enough to maximize model accuracy and 2) to alleviate this, stochastic rounding for model weight updates is one of the minimal support requirement in future model-agnostic training accelerators.

In addition, the theoretical analysis in [3, 4] focuses on rounding for fixed-point numbers, while we study rounding for floating point numbers commonly adopted in accelerators for model-agnostic training. Our analysis reveals that the convergence limit depends on the magnitude of the optimal solution which can be arbitrarily large. This can be a fundamental barrier for convergence which does not emerge in training with fixed point numbers. As suggested by R1 and R2, we will incorporate this discussion into Section 3.1 and related work.

---

> ### Author Response · Authors · 2020-11-18
> **General response to shared comments (2/2)**
>
> **Efficiency of stochastic rounding**: As suggested by R1, R3 and R4, we will elaborate existing efficient implementations of stochastic rounding in Section 3.2 to clarify the hardware efficiency. Specifically in recent literature [5], stochastic rounding for model weight updates can be implemented in an inexpensive way; it does not require any expensive multiplication or division in modern hardware design. Instead, it only requires 1) generating random bit sequence with a shift register [6], 2) add random sequence to the lower mantissa bits and 3) truncate; these operations are inexpensive relative to other optimizer operations regardless of the model size. Second, we note that the minimal technique in our study only requires stochastic rounding for model weight updates while even cheaper nearest rounding remains for forward, backward and optimizer operations other than weight updates.
>
> **Speed and memory efficiency of 16-bit training with Kahan summation**: Despite the fact that 16-bit Kahan accumulation requires an additional 16-bit auxiliary value, it still has advantages over 32-bit and mixed precision training in terms of speed and memory efficiency. In more details, optimizers in both 32-bit and mixed precision training operate on 32-bit weights (for mixed precision, the weights here refer to the master copy) and 32-bit optimizer states (such as momentum). On the other hand, in 16-bit training with Kahan summation, the optimizers leverage fully 16-bit weights, optimizer states and auxiliary variables.
>
> *Regarding the speed*, optimizers in 32-bit and mixed precision training require compute units with 32-bit multiply, while optimizers in 16-bit training with Kahan summation can leverage units with 16-bit multiply. Given that 2) modern MAC units with 16-bit multiply can be implemented with 1.5X throughput compared to those with 32-bit multiply [10] and 3) Kahan summation only introduce 3 additional add/subtract operations which are inexpensive relative to multiply (e.g. Adam has 9 major multiply operations), optimizers in 16-bit training with Kahan summation can demonstrate meaningful speedup.
>
> *Regarding the memory efficiency*, if we use Adam optimizer as an example, 16-bit training with Kahan summation costs 33% and 43% lower memory for weights plus optimizer states respectively than 32-bit training and mixed precision training (mixed precision training has both 16-bit and 32-bit weights in memory). We note that these benefits here are especially pronounced for training large SOTA models such as Megatron [11], Microsoft Zero[12] with billions of model weights using sophisticated optimizers such as Adam with many multiply operations. Following R1, R3 and R4’s suggestion, we will add the above discussion in Section 3.2 to clarify the system efficiency advantage of 16-bit training with Kahan summation.
>
> **Reference**
>
> [1] Hybrid 8-bit Floating Point (HFP8) Training and Inference for Deep Neural Networks, Sun, Xiao, Jungwook Choi, Chia-Yu Chen, Naigang Wang, Swagath Venkataramani, Vijayalakshmi Viji Srinivasan, Xiaodong Cui, Wei Zhang, and Kailash Gopalakrishnan, 2019
>
> [2] Training Deep Neural Networks with 8-bit Floating Point Numbers, Wang, Naigang, Jungwook Choi, Daniel Brand, Chia-Yu Chen, and Kailash Gopalakrishnan, 2018
>
> [3] Analysis of Quantized Models, Hou, Lu, Ruiliang Zhang, and James T. Kwok, 2018
>
> [4] Training Quantized Nets: A Deeper Understanding, Li, Hao, Soham De, Zheng Xu, Christoph Studer, Hanan Samet, and Tom Goldstein, 2017
>
> [5] Understanding and Optimizing Asynchronous Low-Precision Stochastic Gradient Descent, De Sa, Christopher, Matthew Feldman, Christopher Ré, and Kunle Olukotun, 2017
>
> [6] Cloud TPU: Codesigning Architecture and Infrastructure (https://www.hotchips.org/hc31/HC31_T3_Cloud_TPU_Codesign.pdf), Clifford Chao Brennan Saeta, 2019
>
> [7] BFloat16 Processing for Neural Networks on Armv8-A (https://community.arm.com/developer/ip-products/processors/b/ml-ip-blog/posts/bfloat16-processing-for-neural-networks-on-armv8_2d00_a), Nigel Stephens, 2019
>
> [8] NVIDIA Tensor Core Programmability Performance & Precision, Markidis, Stefano, Steven Wei Der Chien, Erwin Laure, Ivy Bo Peng, and Jeffrey S. Vetter, 2018
>
> [9] QPyTorch: A Low-Precision Arithmetic Simulation Framework, Zhang, Tianyi, Zhiqiu Lin, Guandao Yang, and Christopher De Sa, 2019
>
> [10] FPU Generator for Design Space Exploration, Galal, Sameh, Ofer Shacham, John S. Brunhaver II, Jing Pu, Artem Vassiliev, and Mark Horowitz, 2013
>
> [11] Megatron-LM: Training Multi-Billion Parameter Language Models Using Model Parallelism, Shoeybi, Mohammad, Mostofa Patwary, Raul Puri, Patrick LeGresley, Jared Casper, and Bryan Catanzaro, 2019
>
> [12] Turing-NLG: A 17-billion-parameter Language Model by Microsoft, 2020

---

### Author Response · Authors · 2020-11-24
**Draft Revision Uploaded**

Dear reviewers,

Thanks again for all the thoughtful and detailed comments on our initial submission draft.

We have uploaded a revised version of our draft for your further review and reference.

In this updated draft, we incorporated the comments and answer the questions in both the main paper and the appendix. This uploaded version has all the planned edits we mentioned in the review replies (The updated text is marked by blue fonts for convenience.).

Thanks,

Authors of the submission.

---

### Decision · Program_Chairs · 2021-01-07
**Final Decision**

**Decision:**

Reject

**Comment:**

After reading the paper, reviews and authors’ feedback. The meta-reviewer agrees with reviewers that the paper has limited novelty and could be more clear about mix precision training. Therefore this paper is rejected.

Thank you for submitting the paper to ICLR.